# The proliferation of antibiotic resistance genes (ARGs) and microbial communities in industrial wastewater treatment plant treating N,N-dimethylformamide (DMF) by AAO process

Xuan Gao[1], Longhui Xu[1], Tao Zhong[2,3], Xinxin Song[2,3], Hong Zhang[1], Xiaohui Liu[1], Yongbin Jiang[2,3]*

1 Anhui Provincial Key Laboratory of Molecular Enzymology and Mechanism of Major Diseases and Key Laboratory of Biomedicine in Gene Diseases and Health of Anhui Higher Education Institutes, Anhui Normal University, Wuhu, Anhui, China, 2 Department of Environmental Science and Engineering, Anhui University of Technology, Ma'anshan, China, 3 Engineering Research Center of Biofilm Water Purification and Utilization Technology, Ministry of Education, Anhui University of Technology, Ma'anshan, China

* yongbin_jiang@163.com

**Data Availability Statement:** All sequencing files are available from the NCBI database (accession number PRJNA914901).

## Abstract

The excessive use of antibiotics has resulted in the contamination of the environment with antibiotic resistance genes (ARGs), posing a significant threat to public health. Wastewater treatment plants (WWTPs) are known to be reservoirs of ARGs and considered to be hotspots for horizontal gene transfer (HGT) between bacterial communities. However, most studies focused on the distribution and dissemination of ARGs in hospital and urban WWTPs, and little is known about their fate in industrial WWTPs. In this study, collected the 15 wastewater samples containing N,N-dimethylformamide (DMF) from five stages of the anaerobic anoxic aerobic (AAO) process in an industrial WWTPs. The findings revealed a stepwise decrease in DMF and chemical oxygen demand (COD) content with the progression of treatment. However, the number and abundances of ARGs increase in the effluents of biological treatments. Furthermore, the residues of DMF and the treatment process altered the structure of the bacterial community. The correlation analysis indicated that the shift in bacterial community structures might be the main driver for the dynamics change of ARGs. Interestingly, observed that the AAO process may acted as a microbial source and increased the total abundance of ARGs instead of attenuating it. Additionally, found that non-pathogenic bacteria had higher ARGs abundance than pathogenic bacteria in effluents. The study provides insights into the microbial community structure and the mechanisms that drive the variation in ARGs abundance in industrial WWTPs.

## Introduction

ARGs have become a pressing topic in recent times and have been identified as one of the six emerging contaminants by the UNEP [1]. The widespread use of antibiotics in hospital and

**Funding:** We thank the financial supported by the Open Project of Engineering Research Center of Bio film Water Purification and Utilization Technology of Ministry of Education (BWPU2021ZY02) and Outstanding Innovative Research Team for Molecular Enzymology and Detection in Anhui Provincial Universities (2022AH010012). The funders provided support for the sample collection and sequencing work conducted in this study.

**Competing interests:** The authors have declared that no competing interests exist.

agriculture has led to a rapid increase in the number and abundance of antibiotic resistance genes in recent years. However, only 15% of the antibiotics is absorbed and metabolized [2], majority of residual antibiotics being excreted into environment through wastewater. This, in turn, may accelerate the emergence of ARGs in WWTPs. As the largest antibiotic producing and consuming country [3], China is particularly susceptible to higher ARGs risks, which has attracted intensive attention due to their potential health risks.

Growing evidence shows that WWTPs is one of hotspots of ARGs due to various substances are collected in wastewater. It is important to reveal the fate of harmful substance in wastewater treatment systems and their association with ARGs. Although many studies have documented the prevalence of ARGs in WWTPs, most of them mainly focused on the wastewater come from home and hospital [4]. Additionally, Retirement homes was identified to be one of the greatest quantitative hotspots for frequently as well as critically occurring ARGs [5]. However, little is known about the abundance, diversity and transmission of ARGs in industrial wastewater. There is little overlap between the microbial community compositions and ARGs of different WWTPs because of the complexities of wastewater source [6]. Therefore, it is urgently needed to discover the impact of industrial wastewater on ARGs distribution and spread.

DMF is an compound that was synthesized by human and not occur in nature. It has been widely used in diverse chemical, pharma industries because of its excellent miscibility with most organic liquids and water. As one of the components of industrial wastewater, DMF may enter the water body through sewage discharge, and have toxic effects on aquatic organisms, resulting in changes in the ecological structure and function of aquatic communities [7]. The presence of DMF may inhibit the growth and activity of soil microorganisms and destroy the function and stability of soil ecosystem [8]. It is also toxic and can be harmful to the human body, and long-term exposure to high concentrations of DMF may cause health problems, such as skin irritation, respiratory irritation, liver and kidney damage [9]. The terrible thing is that wastewater discharged from the industry is often contaminated with high DMF concentrations, which is environmentally detrimental. Several DMF degrading including Alcaligenes [10], Paracoccus [11] and Methyl bacteria [12] have been identified that are capable of degrading DMF. it is common to introduce these bacteria into WWPTs to degrade the DMF [13]. Among the biological wastewater treatment processes, AAO process has been widely used in industrial WWTPs due to its excellent denitrification removal performance [14]. However, the introduced bacteria may be the major driver impacting the ARGs profile. Although biological treatment could reduce DMF significantly, the abundance of ARGs in the effluents maybe high. Effluents may favor the persistence and spread of antibiotic resistance in the microbial communities of the receiving environments.

Mobile genetic elements (MGEs) are fragments of DNA that can move around the genome, including transposons, integrons, and plasmids. They are capable of horizontal gene transfer between the genetic material of the bacteria, thereby rapidly spreading antibiotic resistance genes through the bacterial population [15]. The spread of MGEs poses a challenge to antibiotic treatment. Because they enable bacteria to quickly acquire antibiotic resistance genes, which can cause antibiotics to become ineffective [16]. In addition, because MGEs are highly plastic and adaptable, they can also promote the evolution and diversification of antibiotic resistance genes [17]. This means that antibiotic use can select not only antibiotic-resistant bacteria, but also bacteria that carry MGEs, further exacerbating the antibiotic resistance problem. Therefore, understanding the effect of MGEs on antibiotic resistance genes is of great significance for formulating rational antibiotic use strategies and antibiotic resistance prevention and control.

This study aimed to investigate the effects of DMF on the distribution of ARGs during the AAO process in industrial WWTPs. It also examined changes in bacterial community

structures and ARGs occurrence during the process, as well as the relationships between ARGs and bacterial genera, specifically pathogens. The results showed that DMF content decreased during the AAO process, while the abundance and number of ARGs and bacterial hosts increased. The study also found that the potential transfer of ARGs in pathogenic bacteria decreased after the AAO process. However, some ARGs still pose a significant threat to public health through their distribution and accumulation in non-pathogenic bacteria, which can be discharged through effluent. Further measures may be necessary to address this issue.

## Method

### Sample collection and DNA extraction

In December 2021, wastewater was sampled from the Shen Nuobei industrial wastewater treatment plant located in Ma'anshan city, Anhui province, China(118.507˚ N, 31.689˚ E). This WWTPs mainly received wastewater containing DMF and operate via anaerobic/anxic/oxic (AAO) techniques. Data on in situ environmental parameters (COD and content of DMF) were provided by the WWTPs. Two liters of wastewater samples were collected from the influent, anaerobic, anoxic, aerobic, and effluent tanks. each sample was collected three times from different locations concurrently. A total of 15 samples were collected (Influent wastewater, IN; Anaerobic process wastewater, AP; Anoxic process wastewater, ANP; Aerobic process wastewater, AEP; Effluent wastewater, EF). the wastewater samples were collected using sampler, immediately transferred to lab and flited with filter membrane. All 0.22 um filter membranes were stored at -20˚C for DNA extraction. The filter membrane was cut into pieces and rinsed with sterile water and used for DNA extraction. DNA extraction for wastewater was performed using FastDNA Spin Kit following the manufacturer's instruction. The extracted DNA samples were assessed by electrophoresis and spectrophotometry to determine their purity. The experimental analysis process is shown in the graphical abstract (S1 Fig).

### Determination and analysis of DMF content

The analysis was carried out by gas chromatography-mass spectrometry. DMF in the sample to be tested was transferred to a suitable organic solvent, and DMF solution with a certain concentration was prepared by dissolution. Prepare a standard solution of a range of DMF concentrations, covering the range of concentrations that may occur in the sample to be tested. The sample to be measured and the standard solution are injected into the gas chromatograph, and the gas chromatograph column and mobile phase are selected for analysis. The qualitative identification was carried out by comparing with DMF mass spectrum. The peak area of DMF in the sample was compared with that on the standard curve for quantitative analysis. According to the relationship between the peak area of each concentration point on the standard curve and DMF concentration, the linear regression analysis was carried out to calculate the DMF concentration in the sample to be measured.

### ARGs quantification using high-throughput quantitative PCR

Targeting ARGs were identified by 294 primer sets, detailed information of the primers are listed in S1 Table. The abundance of ARGs were characterize by high throughput quantitative PCR (HT-qPCR). HT-qPCR was performed using the Wafergen SmartChip Real-time PCR system (Wafergen, Fremont, CA). This SmartChip platform can be used for large-scale analysis by processing 5184-nanowell reactions per run. Absolute gene abundance was calculated as gene copies per litre of wastewater according to Ct (Ct<35) and standard curves. the relative

abundance of genes were determined by normalizing the absolute copy number of ARGs to the copies of 16S rRNA gene.

### Illumina sequencing, data processing and analysis

While the active bacterial community was profiled by amplicon sequencing the 16S rRNA gene. To assess the diversity and relative abundance of bacteria in all samples, the V3 and V4 region of 16S rRNA was amplified. The initial enzyme activation was performed at 95 ˚C for 5 min, and then 35 cycles of the following program were used for amplification: 95 ˚C for 30 s, 58 ˚C for 30 s and 72 ˚C for 30 s. By sequenced with Illumina Hiseq 2500 platform universal primers 341F and 806R by Yuanzai biotechnology Co., Ltd (Hefei, China). The raw pair-end reads were assembled after filtering of adaptor, low-quality reads, ambiguous nucleotides, and barcodes to generate clean joined reads capturing the complete V4-V5 region of the 16S rRNA gene. Quantitative Insights Into Microbial Ecology (QIIME) was used for further data processing [18]. The raw reads have been submitted to NCBI with the SRA database accession of https://www.ncbi.nlm.nih.gov/bioproject/PRJNA914901

### Correlations analysis between bacterial community and ARGs

The possible potential hosts (at the phylum level) of ARGs and the relationships among them were constructed by correlation analysis. Furthermore, co-occurrence network analysis was used to showing the associations among ARGs and pathogen (at the genus level). In order to filter the data for reduced network complexity, the connections shown here stand only for strong (Spearman's r >0.7) and remarkably significant (p< 0.01) correlation. Spearman correlations among ARGs and bacterial community were determined using the "psych" package in R(v4.1.2). The network analysis was constructed using "vegan" and "hmisc" packages in the R. Visualization of network analysis was produced on the Gephi software (v0.9.2).

### Data visualization and statistical analysis

The average and standard deviations of the abundances for each ARGs, MGEs, and 16S rRNA were determined using R software. To visualize the similarity and dissimilarity in the ARGs and microbial communities among different samples, principal component analysis (PCA) analysis were performed based on different ARGs and OTUs. Heatmap plotting and venn charts created using R package "pheatmap" to show the fate of ARGs and bacterial community.

## Results

### Dynamic changes of chemical parameters and PCA analysis in industrial WWTPs

To test the dynamic of nutrient removal efficiency, investigated wastewater physicochemical properties (DMF and COD) during the whole stage of AAO process. Removal efficiency of DMF was 99.6%. The content of DMF decreased from 476.3 mg/L to 1.8 mg/L (Fig 1A) while the value of COD decreased from 38049 mg/L to 79.4mg/L (Fig 1B) during AAO process. Therefore, the content of COD was consistent with that of DMF. There was a significant positive correlation (Pearson test, p = 7.003e-16) between DMF and COD (Fig 1C). The abundance of 16S rRNA was significantly higher in AP, ANP, AEP and EF stage than in IN (Fig 1D). The copy number of 16S RNA measured at the IN was 8,114, while the value at the EF increased to 42,706, which suggesting that a lot of bacterium may involve in the process of degeneration of DMF.

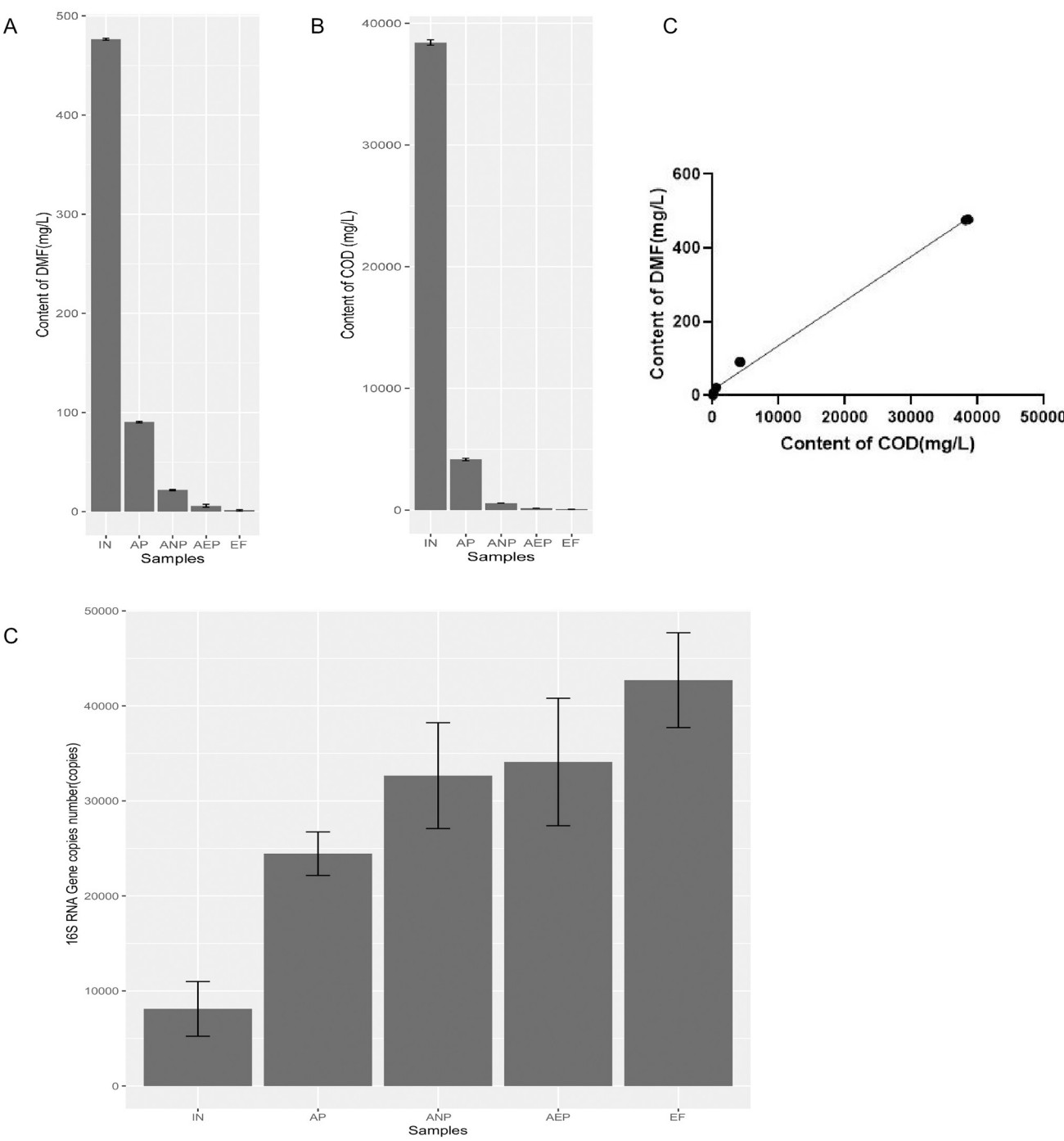

**Fig 1. The change among the different treatment in WWTPs.** (**A**) Dynamic changes of the content of DMF and (**B**) COD; (**C**) Correlation analysis of DMF and COD; (**D**) The abundance of 16S rRNA; Influent wastewater, IN; Anaerobic process wastewater, AP; Anoxic process wastewater, ANP; Aerobic process wastewater, AEP; Effluent wastewater, EF.

## The abundance and distribution of ARGs in WWTPs

The PCA plot showed that ARGs from the same treatments were clustered together, exhibiting similar profiles (Fig 2A). Further analysis of data demonstrated that the ARGs distribution of IN samples was distinctly different from the other treatment samples along the PCA axis (explaining 57.31 of variations). These results suggest that the abundance and diversity of ARGs were significantly changed between the IN samples and treatments. In addition, the replicated samples of ANP, AEP, and EF samples were clustered while that of IN is obviously separated which indicated that the ARGs abundances were more stable in treatments than in IN.

Differentially relative abundant ARGs found in wastewater during the AAO process are displayed in Fig 2B, with AP being the highest. The application of activated sludge in AP, ANP, and AEP pool reduced the content of DMF (Fig 1A), however, it also significantly increased the absolute abundance of ARGs as well as MGEs leading to a great enrichment than IN (Fig 2C). Calculated the fold enrichment of ARGs in AP relative to that in IN, which ranged from 3 to 5 times. *MexF*, *tna-04*, and *qacEdelta1-02* were identified as the top 3 dominant ARGs in IN. Multidrug, transposase and sulfonamide were the dominant ARGs in AP, ANP, and AEP (Fig 2D). Noteworthily, the abundance of 16s RNA dramatically increased during AAO process (Fig 1C). Infer that the increase abundance of ARGs in WWTPs may be related to bacterium community, because many previous studies have demonstrated that some phylum of strain was closely related to the enrichment of ARGs [19, 20]. These results suggest that the AAO process acted as a microbial source increased rather than attenuated the total abundance of ARGs. The number of ARGs detected varied significant during AAO process. Among these samples, the number of ARGs was highest in AP and lowest in IN. However, the resistance mechanisms of ARGs detected were similar, with antibiotic deactivation being the dominant resistance mechanism in all samples (Fig 2E).

## The changes of ARGs during the AAO process in WWTPs

To inverstigate the fate of ARGs, venn diagram was performed to show the distribution of ARGs in wastewater treatment process. The number of unique ARGs between IN, AP, ANP, AEP, and EF samples account for 8%, 0%,2%, 3%, and 2% of total ARGs in each of these group, respectively. Ten unique ARGs were detected in IN indicated that these ARGs can be eliminated totally (Fig 3A). These ARGs including *aac(6')-II*, *aadA-1-01*, *aadA-1-02*, *blaIMP-01*, *blaPER*, *cmr*, *cmx(A)*, *dfrA1*, *mexE*, *tnpA-03*. There were 53 genes shared between the IN and treatment wastewater, indicating that these ARGs may not be eliminated by AAO process (Fig 3B). 18 new ARGs will generate during the wastewater treatment process demonstrated that the active sludge may be the reservoirs for ARGs (Fig 3C). As the heatmap illustrated, most of the 53 share ARGs were decreased while 18 new generate ARGs were increased. These results indicated that bacteria in sludge from AP, ANP, and AEP rather than the wastewater from IN is the main source of ARGs in WWPT.

## Bacterial abundance and size distribution

The PCA of bacterial communities and ARGs profiles all revealed that the samples were clustered by sample type and exhibiting similar profiles. In addition, the abundance of total ARGs were increased as the copy number of 16S RNA during the wastewater treatment (Fig 4A). These results indicated that the vertical gene transfer and horizontal gene transfer caused by reproduction of bacterial host were the main modes of ARGs proliferations in WWTPs. Proteobacteria was the dominant phyla in all the stage. The composition of Firmicutes decreased form 35% in IN to 0.57% in EF while Chloroflexi increased from 2.46% to 11.44%. with the DMF concentration decreasing from 476.3 mg/L to 1.8 mg/L, the composition of Bacteroidetes

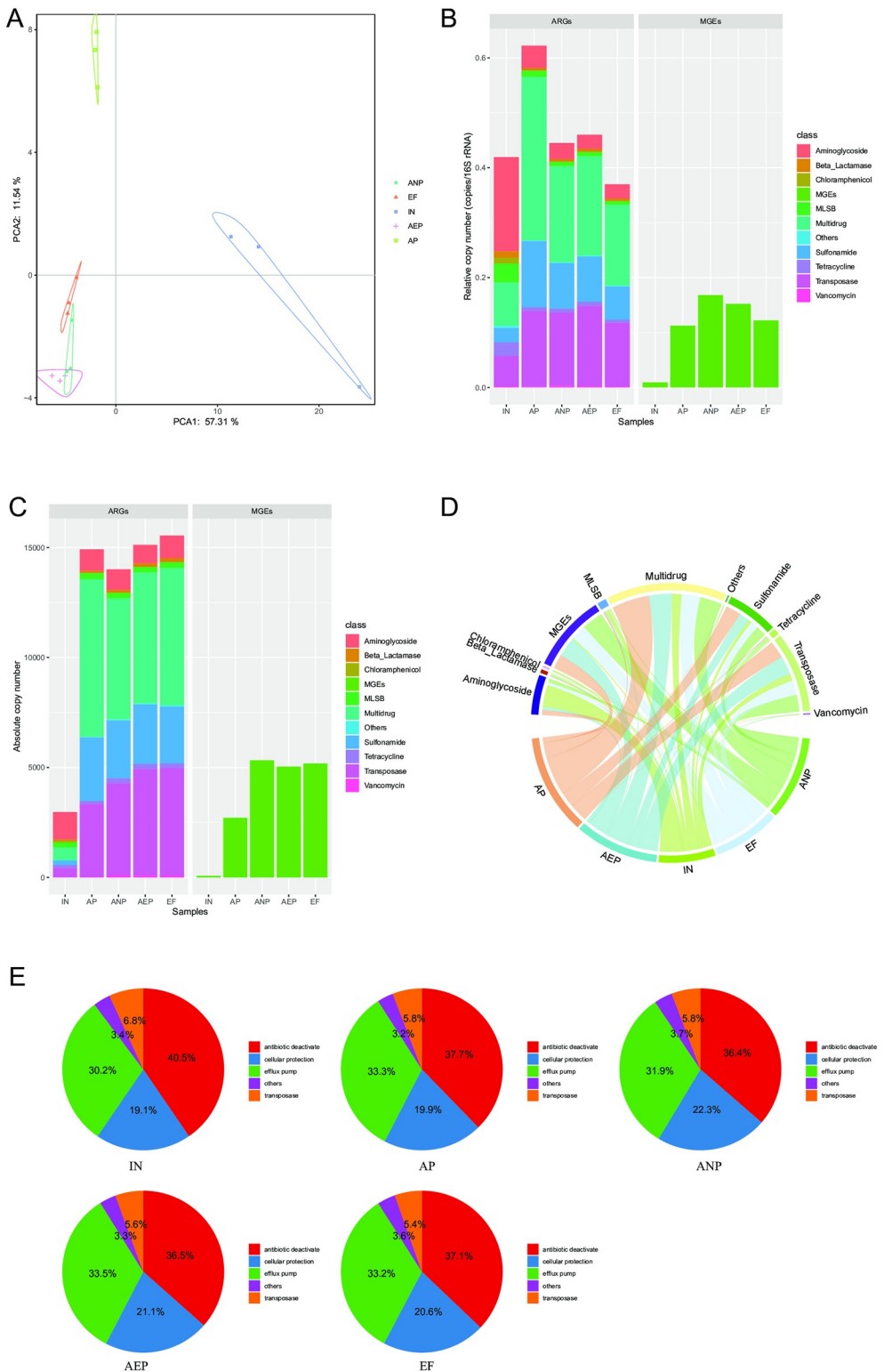

**Fig 2. The abundance of ARGs in WWTPs.** (**A**) PCA analysis between different samples in WWTPs. (**B**) the relative abundance of ARGs in WWTPs; (**C**) the absolute abundance of ARGs in WWTPs; (**D**) Chord diagram analysis between the sample and ARG subtype; (**E**) the mechanisms of ARGs variation in WWTPs.

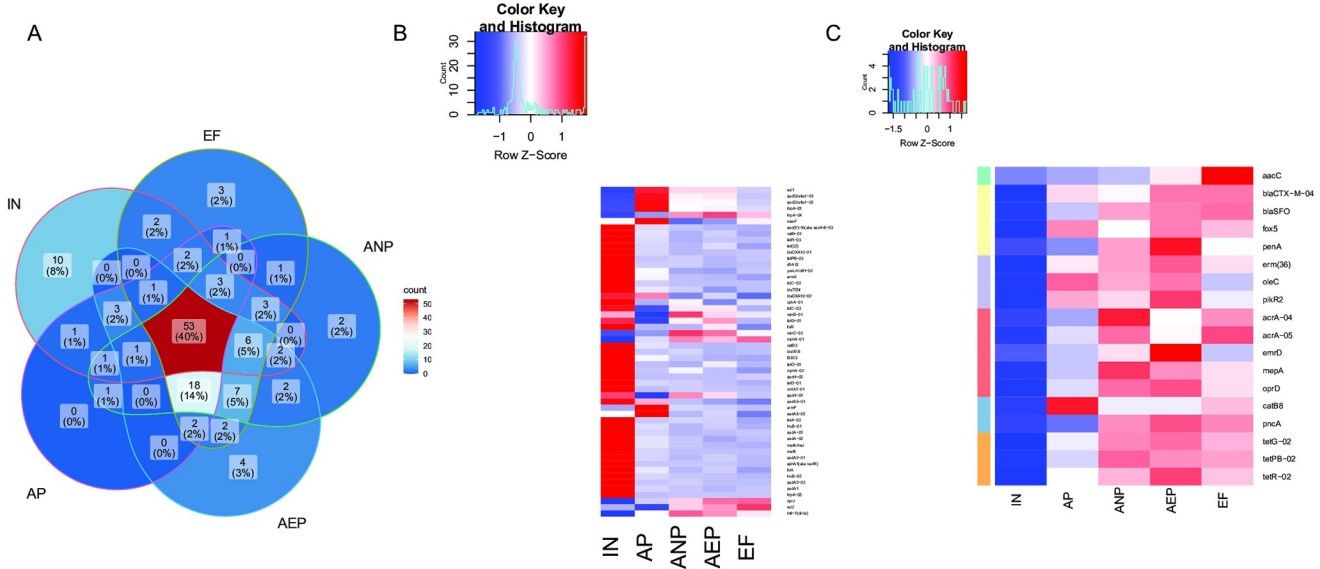

**Fig 3. The ARGs changed along the WWTPs process.** (**A**) venn diagrams analysis between the samples. (**B**) 53 share genes changed between samples; (**C**) 18 new generated ARGs changed between samples.

and Acidobacteria gradually increased from 0.32% to 10.31% and 0.42% to 3.81%, respectively (Fig 4B). These results showed that some bacteria belonging to the Bacteroidetes and Acidobacteria phylum were sensitive to DMF.

In the present study, the relative percentages of ARB and ARGs subtypes at each stage of WWTPs appeared to be similar and a dramatic increase in ARGs occurred during AAO process. The relative abundance of some phylum, such as Acidobacteria, Latescribacteria, and Gemmatimonadetes were significantly increased through AAO process. On the contrary, some phylum, such as Firmicutes, Deinococcus, and Cyanobacteria were decreased (Fig 4C). The spread and distribution of 13 pathogens are presented in Fig 4D. The abundance of some pathogens, such as Legionella, increased after water treatment. However, most genera belonging to pathogen were removed after the wastewater treatment. These results suggest that ARB population shift toward nonpathogenic ARGs carriers during AAO process.

## The correlation between ARGs and their host bacteria

Log2 transformed of fold change was used to describe ARGs from influent to effluent. Some biggest changed ARGs abundance was calculated as the log of the ratio between IN and EF, which ranged from -2.5~7. *OprJ*, *intI-1*, *sul*, *qacEdelta1-01*, *qacEdelta1-02* showed high fold changes. To identify entire ARGs profile drivers in the dataset, performed the correlated analysis between entire ARGs and bacterial composition. As shown in Fig 5A, Deinococcus and Firmicutes phyla were identified as the dominant potential hosts for various ARGs and MGEs. *ErmF*, *addA5-02*, and *intI-01* were observed to widely coexist with bacterial community which indicated that these ARGs may easily spread in WWTPs. The potential host bacteria of the *oprJ* were Acidobacteria. ARGs became increasingly more correlated with the presence of bacterial, which were primarily identified as the Proteobacteria, Actinobacteria. Interestingly, the potential ARGs host bacteria included Actinobacteria with *qacEdelta1* and Proteobacteria with *intI-1*. Overall, Firmicutes, Deinococcus, Acidobacteria, Actinobacteria, and Proteobacteria

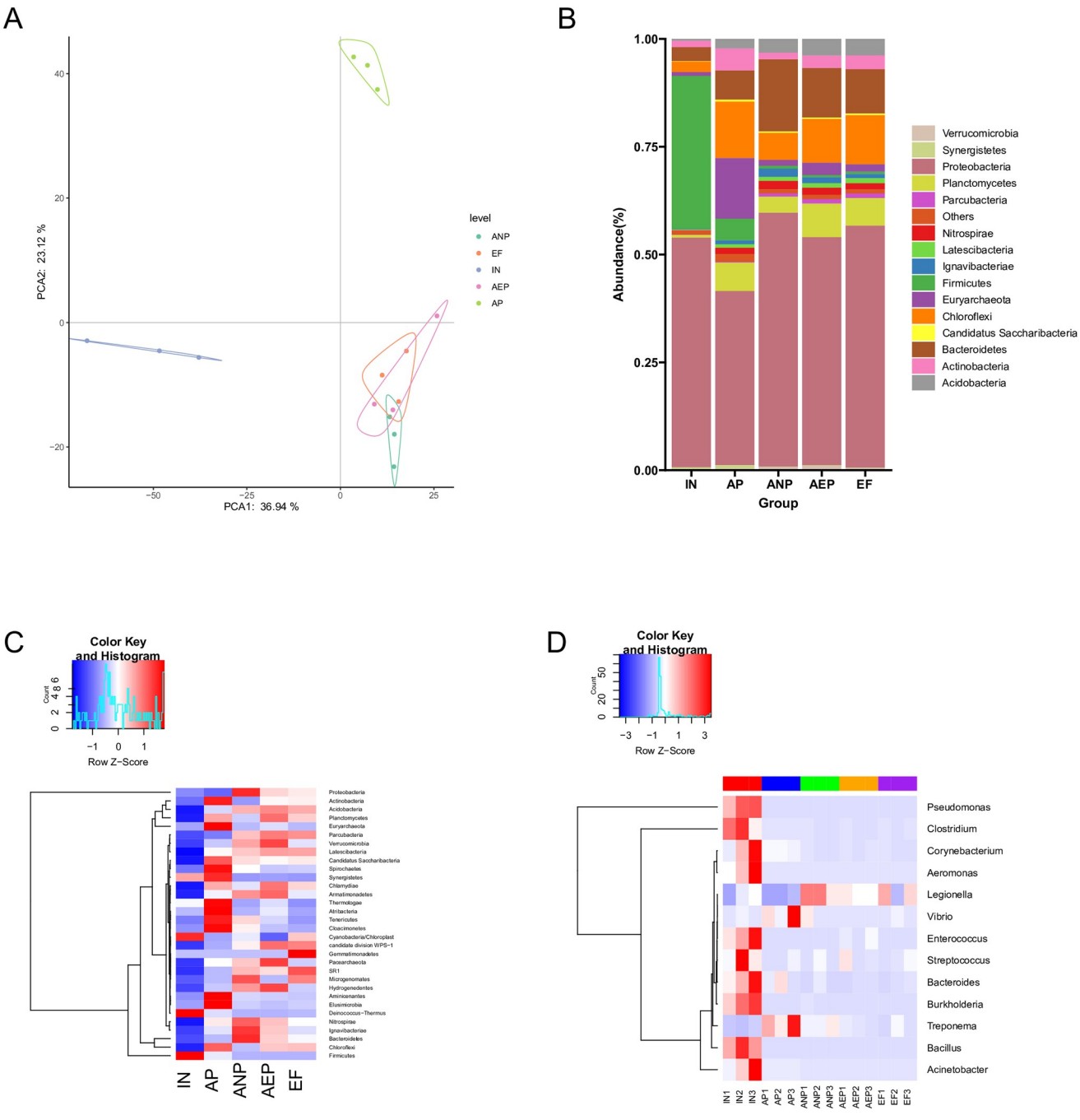

**Fig 4. Changes of bacterial community.** (**A**) PCA analysis between samples base on the OTUs; (**B**) the structure of bacterial between samples; (**C**) the variation of bacterial at phylum level between samples. (**D**) the variation of pathogen at species level between samples.

had the strongest correlates among the entire ARGs profiles. These results shown that these four dominant bacteria phyla may be the main potential host of ARGs.

To find out the mechanism causing the variation of ARGs transfer between pathogen, network analysis was adopted to quantify the ARGs contribution to pathogen bacteria and identify the potential transmission pathway. the relationship among possible potential pathogen host at

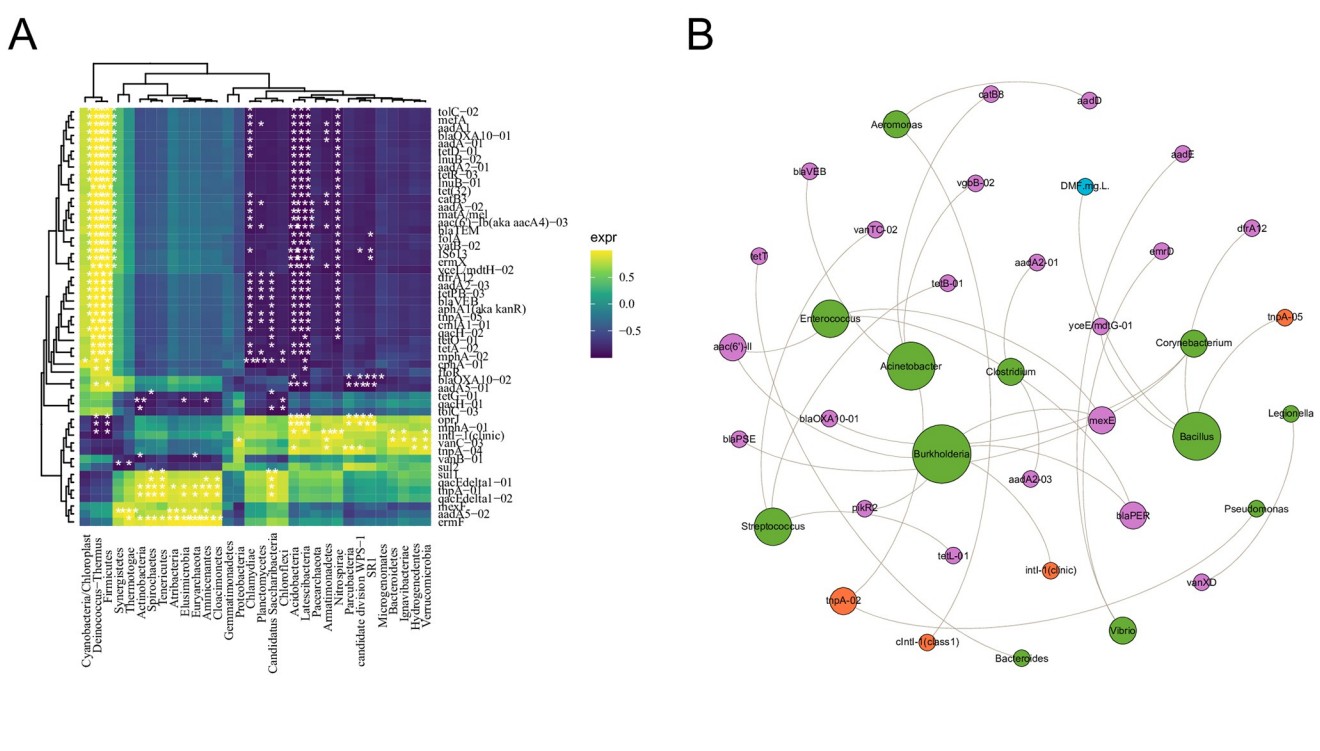

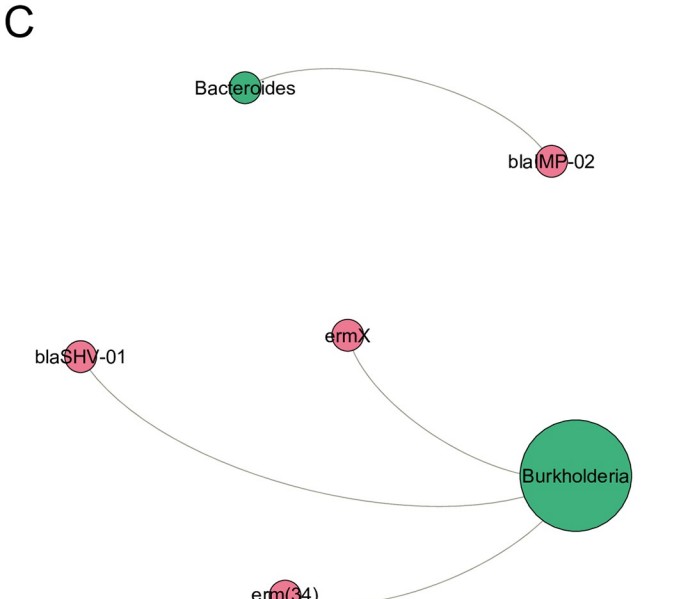

**Fig 5. The correlation between ARG and their host bacteria.** (**A**) the analysis base on Spearman's correlation between ARGs and their host; network analysis revealing the correlations among the fates of ARGs, MGEs, DMF and bacterial in influent (**B**) and effluent (**C**).

the genera level, ARGs, MGEs, and DMF were constructed by network analysis from influent and effluent. Co-occurrence patterns of the ARGs and potential pathogen host was revealed. As shown in Fig 5B, 12 genera were identified as the pathogen host for ARGs, MGEs, and DMF in influent. Most identified bacteria genus, which belonged to Firmicutes, Proteobacteria, and

Bacteroidetes were related to the ARGs and MGEs, such as *metXE*, *blaOXA10-1*, *pikR2*, *intI*, *tnpA*, and so on. Specifically, DMF had a significant positive correlation with Bacillus, which was belonging to Firmicutes, indicated that DMF may make Bacillus a competitive advantage by inhibiting the growth of other strains. Furthermore, found a strong correlation between Burkholderia and *intI*, suggesting the potential of horizontal gene transfer (HGT) in influent. Notably, the potential transfer of ARGs in bacterial pathogens decreased after wastewater treatment. As shown in Fig 5C, only Burkholderia and Bacteroides were detected in the effluent. Similarly, the number of ARGs in pathogens decreased. *BlaSHV-01*, *ermX*, and *erm* were still detected in effluent. According to the enrichment folds of ARGs in influent compared with that in effluent, most of the ARGs could be easily eliminated. However, ARGs belonging to Beta_lactamase and MLSB were hard to eliminate across the pathogens.

## Discussion

### The correlation between ARGs and the composition of wastewater

As a newly emerging environmental pollution, ARGs have increased rapidly and threats to human health [21]. WWTPs are potential hotspots for the spread of antibiotic resistance and transfer of ARGs. Therefore, great attention was paid to the distribution of ARGs in WWTPs. Previous research reported that the wastewater from hospital containing a large number of antibiotics was fed into WWTPs, leading to a significant increase in some ARGs, such as *blaCTX*, *blaTEM*, and *qnrS* [22]. Besides the hospital wastewater, many researches focus on WWTPs which received wastewater from municipal, pharmaceutical enterprises, poultry and livestock farm.

There is little overlap between the ARGs composition of different WWTPs due to the complexities of wastewater sources. In the study, the DMF and COD concentration at IN reached 476mg/L and 38,049mg/L, respectively. Due to the differences in dissolved organic matter (DMF), wastewater quality (higher COD concentration) affects the concentrations and distribution of ARB and ARGs. The main ARGs detected in the influent of municipal WWTPs are the genes resistant sulfonamides (*sul1* and *sul2*) and tetracyclines (*tetA*) [23, 24]. Some ARGs such as *dfrA12*, *dfrA13* [6], *blaTEM* [25], and *intI1* [26] have been predominantly detected in many WWTPs. In the study, the ARGs analysis in the influent of industrial WWTPs revealed the highest prevalence of ARGs related to aminoglycoside, multidrug, and transposase. Some ARGs, such as *mexF*, *tna-04*, and *qacEdeltal-02* are detected as the dominant genes in the influent which is rarely reported in other WWTPs. These results indicated that WWTPs receive different wastewaters, which contain different concentration of substances could have a certain impact on the structure of ARGs. Therefore, it is necessary to explore the dynamics of ARGs in different WWTPs respectively.

### The main drivers of ARGs profiles in industrial WWTPs

In previous study, the physicochemical properties of wastewater (such as heavy metals, natural organics), bacterial communities, and MGEs were possible the main drivers of ARGs profiles. Heavy metals [27] effects the concentrations and distribution of ARGs. Many researchers believe that the bacterial community shift rather than any individual factor was the major driver shaping the relative abundance of ARGs in WWTPs [28]. The abundance of total ARGs and 16S rRNA had significant correlation in the research, which may suggest the bacteria is major driver of ARGs. The results from previous report showed that the dynamics of bacterial community structures induced by changes in environmental attributes may be the main driver for shift in ARGs profiles [29], which was consistent with our results. Some ARGs (*ermF*, *aadA5-02*, and *intI-1*) have a significant correlation with most of genera, indicating the

existence of multiple antibiotic resistance bacteria (ARB). Deinococcus and Firmicutes were the strongest correlation of the entire ARGs profile, suggesting their potential contribution to the generation of ARB during WWTPs processes. Further research demonstrated that Inorganic nutrients, such as nitrate [30], ammonia [31], phosphorus [32] also contribute to the HGT of ARGs between bacteria. Higher concentration of COD in wastewater represent higher natural organics increase the transformation between resistant bacteria [6]. There is growing evidence of the capacity of chemical substance to enhance the propagation and dissemination of ARGs between bacteria via HGT [33]. The stochastic processes of dispersal and drift as well as deterministic factor is considered as important parameters in shaping the microbial community [34]. However, in the research, network analysis revealed that a weak correlation between ARGs and DMF. These results suggested that microbial community structure in industrial WWTPs was the main factor for ARGs evolution rather than other factors.

Many people believe that WWTPs can significantly reduce the number and abundance of ARGs in wastewater, as it has been well-established in studies on domestic and medical wastewater [5]. However, research on industrial indicates that the abundance of ARGs actually increases rather than decreases during wastewater treatment. This is because the influent of industrial wastewater contains large amounts of toxic substances, such as DMF, which are gradually removed during AAO process. At the same time, many microorganisms that degrade toxic substances are artificially added during treatment, providing favorable conditions for the enrichment of ARGs due to the increased diversity and abundance of microorganisms. It is commonly believed that only wastewater containing antibiotics discharged into the environment can lead to an increase in ARGs, which may be overestimated. This study suggests that any wastewater treatment that involves microbial treatment may lead to an increase in ARGs.

## Correlation of ARGs and pathogen

It is essential to evaluating the potential transfer of ARGs in pathogenic bacteria for its exposure risks in human health. Therefore, the reduction and accumulation of pathogenic ARB should be monitored during all stages of WWTPs process. The percent abundance of ARGs in pathogen significantly decreased sharply in most WWTPs [6, 35]. Previous research reported that some pathogens (Bacteroides, Klebsiella, and Salmonella) may harbor more ARGs than nonpathogenic bacteria [6]. In the study, most bacterial species present in effluent are assumed to be nonpathogenic. The industrial WWTPs effectively eliminate pathogenic bacterial communities may providing unfavorable conditions for pathogens. Considering the composition of pathogen decreased along the AAO process, the correlation between pathogen and ARGs may also change, leading to the low pathogen associated ARGs abundance. However, the accumulation of ARGs in nonpathogen in effluent still pose potential risks to the environment and human health.

ARGs in WWTPs exhibit significant different due to the wastewater from different sources. In the study, explore the dynamic change in industrial wastewater containing DMF. Found that bacterial community is the major drivers of ARGs. Though the pathogenic bacteria and related ARGs is decreased, the horizontal gene transfer via natural transformation among nonpathogen remains high. Therefore, WWTPs could reduce the pathogens of receiving environment while the overall risk was not decreased [36, 37]. Ultimately, it is a matter of implementing measures to counteract the spread of ARGs into environment.

## Conclusion

The increase of abundance of ARGs is becoming recognized as an emerging global environmental problem. However, industrial WWTPs as the hotspot of ARGs may be the source were

neglected [38–40]. Biological treatment often used to remove the pollutants with high diversity of bacterial community without considering the spread of ARGs through the HGT between bacteria. In this research, investigated the distribution of ARGs and bacterial community along a typic AAO process in industrial WWTPs. AAO process can remove the DMF efficiently and bacteria plays an important role in this process. Several ARGs belonging to multi-drug, Transposase, and Sulfonamide were significantly increased. The abundance of 16S rRNA were dramatic increased. Approximate 30% new ARGs produced during the treatment. These results indicated that the shift in bacterial communities induced by WWTPs might be the main driver in shaping ARGs. Network analysis can further identify the correlations between ARGs and pathogen. This result indicated that most of pathogen were decreased in effluent and less ARGs were harbored between them. Taken together, beside the removal efficiencies of traditional pollutants should be improved, the changes in ARGs number and abundance should be considered when the WWTPs upgrades are implemented in the future.

## Environmental implication

Antibiotics are widely used in the world, leading to antibiotic resistance gene contamination in the environment, which will pose a great threat to public health. WWTPs are repositories of ARGs and are considered hot spots for HGT between bacterial communities. Biological treatment is usually used to remove contaminants with high diversity in bacterial communities, regardless of the spread of ARGs between bacteria via HGT. In this study, investigated the distribution of ARGs and bacterial communities during typical AAO processes in industrial WWTPs. These findings may provide insight into the microbial community structure and the mechanism transformation of ARGs variation in industrial WWTPs.

## Supporting information

**S1 Fig. Graphical abstract.**
(PDF)

**S1 Table. Primers used for ARGs detection and 16S rRNA sequencing.**
(DOCX)

## Acknowledgments

We thank all the staff at Shen Nuobei industrial wastewater treatment plant in Ma'anshan, Anhui province, China for their support.

## Author Contributions

**Data curation:** Longhui Xu.

**Funding acquisition:** Yongbin Jiang.

**Investigation:** Tao Zhong.

**Methodology:** Xinxin Song.

**Resources:** Xiaohui Liu.

**Software:** Hong Zhang.

**Supervision:** Yongbin Jiang.

**Writing – original draft:** Xuan Gao.

**Writing – review & editing:** Xuan Gao, Yongbin Jiang.

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
