## [Decision Letter · Decision Letter 0]

21 Nov 2023

PONE-D-23-27587The proliferation of antibiotic resistance genes (ARGs) and microbial communities in industrial wastewater treatment plant treating N,N-dimethylformamide (DMF) by AAO processPLOS ONE

Dear Dr. Gao,

Thank you for submitting your manuscript to PLOS ONE. After careful consideration, we feel that it has merit but does not fully meet PLOS ONE’s publication criteria as it currently stands. Therefore, we invite you to submit a revised version of the manuscript that addresses the points raised during the review process.

We look forward to receiving your revised manuscript.

Kind regards,

Catarina Leite Amorim, Ph.D.

Academic Editor

PLOS ONE

“We thank the financial supported by the Open Project of Engineering Research Center of Bio film Water Purification and Utilization Technology of Ministry of Education (BWPU2021ZY02) and Outstanding Innovative Research Team for Molecular Enzymology and Detection in Anhui Provincial Universities (2022AH010012)”

5. Please upload a copy of Supporting Information Table S1 which you refer to in your text on page 3.

Reviewers' comments:

Reviewer's Responses to Questions

**Comments to the Author**

1. Is the manuscript technically sound, and do the data support the conclusions?

Reviewer #1: No

Reviewer #2: Yes

Reviewer #3: Yes

2. Has the statistical analysis been performed appropriately and rigorously? 

Reviewer #1: I Don't Know

Reviewer #2: Yes

Reviewer #3: Yes

3. Have the authors made all data underlying the findings in their manuscript fully available?

Reviewer #1: No

Reviewer #2: Yes

Reviewer #3: Yes

4. Is the manuscript presented in an intelligible fashion and written in standard English?

Reviewer #1: Yes

Reviewer #2: Yes

Reviewer #3: Yes

5. Review Comments to the Author

Reviewer #1: This work uses a very powerful tool to analyze all the resistome in environment samples. The rationale to support analyzing a WWPT degrading the DMF is weak, and the discussion of the results is superficial. The methodology is very general, and many details are missing.

Define AAO acronym in the abstract

Introduction

May the Authors support the statements about introduced bacteria? “it is common to introduce these bacteria into WWPTs to degrade the DMF.” Or “However, the introduced bacteria may be the major driver impacting the ARGs profile. Although the biological treatment could reduce DMF significantly, the abundance of ARGs in the effluents may be high. and effluents may favor the persistence and spread of antibiotic resistance in the microbial communities of the receiving environments.”

The introduction may be more supportive of the relevance of analyzing ARGs in a DMF containing wastewater. Why could this chemical be relevant to ARG or their transferring in the microbial community?

The introduction and final paragraph seem to be a summary of the methodology and results. In the authors’ guidelines, only a conclusive brief statement is required. Please be more specific.

Methodology

Could the authors may be specific about their “samples replicates”?. Do sampling was carried out on different days? Please specify

Line 106. Please justify “The abundance of expressed ARGs”, if the analysis was DNA-based, authors can’t claim “gene expression”. In the same sense, lines 120, 129 mention transcripts. Is it correct?

The methodology for DMF analysis is missing.

Table S1 is missing.

Results

How the 16S rRNA Absolute gene abundance was quantified? This methodology is missing. How do the authors explain a higher abundance in the effluent (Fig 1c)? The authors may explain better the process in the WWTP and their conditions? Does the WWTP have a settling unit for aerobic flocs? The units of 16S rRNA concentration in Figure 1c are missing.

Line 157. Why the different IN clustering was so “obviously”?

Define the MGE acronym.

The authors may support this discussion with a deep literature review: “We infer that the increase abundance of ARGs in WWTPs may be related to bacterium community, because many previous studies have demonstrated that some phylum of strain was closely related to the enrichment of ARGs.”

Figure 2E, is lacking the label corresponding to each pie chart.

How did the authors select the pathogenic bacteria from the whole bacteria community? This methodology is unclear.

The authors may justify this statement “Specifically, DMF had a significant positive correlation with Bacillus, which was belonging to Firmicutes, indicated that DMF may promote the growth of Bacillus. Furthermore”. A positive correlation doesn’t mean a biological activity. This is the only discussion about the effect of DMF.. So, it seems irrelevant to the study, which contradicts the justification in the introduction. Authors may reconsider if DMF can be a determinant parameter to study.

Reviewer #2: 1 First person must not be used in the manuscript, such as our, we.

2 In Abstract, the DMF should be first given the complete name, not only abbreviation.

3 the keywords were not approviate, DMF should be included.

4 The instrument of HT-qPCR should be given.

5 The gene name should be italic.

6 The ARGs should be analyzed by different category.

7 The component of wastewater have an effect on abundance of ARGs this sentence should be changed, for “have an effect” was nonsense.

8 The effect of DMF should be focused.

9 The effect of MGEs should be focused.

Reviewer #3: In this study, wastewater from five stages of the AAO process at a designated water plant was collected and analyzed for COD, DMF, ARGs, and microbial community structure. Results showed that the AAO process may act as a microbial source that increases the total abundance of ARGs, and the wastewater had higher abundance of ARGs for non-pathogenic bacteria than for pathogenic bacteria. Overall, this study provides information on microbial community structure and the mechanisms driving changes in the abundance of ARGs in industrial wastewater treatment plants. However, before considering publication, the following questions should be addressed. My specific comments are as follows:

1. Authors should be aware of spelling and formatting errors, eg. line 67.

2. This study sampled only one specific water plant which is not representative and the conclusions may not be generalizable. The authors should add some related references for comparison such as Journal of Environmental Management,Volume 347, 2023, 119053, Journal of Hazardous Materials,Volume 298, 2015, Pages 303-309, Process Safety and Environmental Protection,Volume 93, 2015, Pages 68-74.

3. COD was consistent with the amount of DMF. A correlation chart is suggested to add to make it clear.

4. Abbreviations need to be spelled out in full the first time they appear, DMF,AAO in the abstract, PCA in the methodology on line 31.MGEs in the result.

5. In the conclusion, it is stated that the DMFs in Figure 2b for AP, ANP, AEP, but the graph shows significantly higher values for AP than for influent and effluent.Please check it.

6. PLOS authors have the option to publish the peer review history of their article (what does this mean?). If published, this will include your full peer review and any attached files.

Reviewer #1: No

Reviewer #2: No

Reviewer #3: No

---

## [Author Response · Author response to Decision Letter 0]

30 Dec 2023

Thank you for taking the time to review our manuscript. We appreciate your constructive feedback, and we have carefully considered your comments. Below are our responses addressing the specific concerns you raised:

Reviewer #1：

1．This work uses a very powerful tool to analyze all the resistome in environment samples. The rationale to support analyzing a WWPT degrading the DMF is weak, and the discussion of the results is superficial. The methodology is very general, and many details are missing.

Reply: Thank you for your valuable feedback. We have taken your suggestions into consideration and have made some enhancements to our submission. Specifically, we have added more details regarding the methods employed in our study as follow: 

“Determination and analysis of DMF content

The analysis was conducted using gas chromatography-mass spectrometry, a highly reliable and precise method. In order to test for the presence of DMF in the sample, it was first transferred to a suitable organic solvent and dissolved to create a DMF solution with a specific concentration. To ensure accurate results, a series of standard solutions with varying concentrations of DMF were prepared. These standard solutions covered a range of concentrations that could potentially be found in the sample under investigation. Both the sample and the standard solutions were then injected into the gas chromatograph. Careful consideration was given to the selection of the gas chromatograph column and mobile phase, ensuring optimal conditions for analysis. For qualitative identification, the mass spectrum of the DMF in the sample was compared to that of a known DMF reference. This comparison allowed for the confident identification of DMF in the sample. Quantitative analysis was carried out by comparing the peak area of DMF in the sample to that of the standard curve. By establishing the relationship between the peak area and the concentration at each point on the standard curve, a linear regression analysis was performed. This analysis enabled the accurate calculation of the DMF concentration in the sample under investigation.”

In the discussion, we provided additional clarification on the relationship between microbial communities, antibiotic resistance genes (ARGs), and dissolved organic matter (DMF). 

“Previous studies (1) have extensively investigated a total of 20,762 metagenomes, revealing the presence of DMF-degrading bacteria in 952 genomes. These bacteria, predominantly belonging to the Rhizobiales, Paracoccus, Rhodococcus, Achromobacter, and Pseudomonas genera, are primarily aerobic and are found within the Proteobacteria and Actinobacteria phyla. Intriguingly, these two phyla are also known to be enriched with antibiotic resistance genes (ARGs). Based on these findings, we propose a hypothesis that suggests that these bacteria play a crucial role in the degradation of DMF during sewage treatment. However, an unintended consequence of their presence may be the increased enrichment of ARGs, thereby elevating the abundance of these genes in the treated wastewater. While this assumption is currently based solely on experimental results, it presents a compelling argument suggesting that the resistant bacteria community may facilitate the reduction of DMF levels in sewage.”

2．Define AAO acronym in the abstract

Reply: Thanks for your advice. Adjustments have been made to follow your suggestions, and abbreviations are spelled out in full when they first appear.

3．May the Authors support the statements about introduced bacteria? “it is common to introduce these bacteria into WWPTs to degrade the DMF.” Or “However, the introduced bacteria may be the major driver impacting the ARGs profile. Although the biological treatment could reduce DMF significantly, the abundance of ARGs in the effluents may be high. and effluents may favor the persistence and spread of antibiotic resistance in the microbial communities of the receiving environments.”

Reply：Thank you for your thoughtful consideration of our manuscript. We acknowledge the importance of providing adequate support for statements regarding the introduction of bacteria into wastewater treatment plants (WWPTs) for DMF degradation. In the revised manuscript, we will include additional references and data to substantiate the claim that it is common practice to introduce specific bacteria for the purpose of DMF degradation. This will enhance the credibility and robustness of the statements in question. “it is common to introduce these bacteria into WWPTs to degrade the DMF (2).” “However, the introduced bacteria may be the major driver impacting the ARGs profile. Although the biological treatment could reduce DMF significantly, the abundance of ARGs in the effluents may be high. and effluents may favor the persistence and spread of antibiotic resistance in the microbial communities of the receiving environments,” A recent study has shown that effluents from wastewater treatment plants are a significant source of antibiotic-resistant bacteria and antibiotic-resistant genes (ARGs), posing a significant risk to public health. Therefore, We propose a hypothesis that the treatment of DMF, by introducing degradation bacteria, may lead to the dissemination and spread of antibiotic resistance genes (ARGs).

4．The introduction may be more supportive of the relevance of analyzing ARGs in a DMF containing wastewater. Why could this chemical be relevant to ARG or their transferring in the microbial community?

Reply: Thank you for your thoughtful consideration of our manuscript. The existence of ARGs in microorganisms may be related to environmental stress, such as metal can promote microbial strains to produce resistance, which is related to the co-evolution of genes. As a commonly used chemical raw material, DMF may promote the existence or transfer of some ARGs in strains through this co-evolution in the environment.

5．The introduction and final paragraph seem to be a summary of the methodology and results. In the authors’ guidelines, only a conclusive brief statement is required. Please be more specific.

Reply: Thank you for your thoughtful consideration of our manuscript. The summary description of the last paragraph of the introduction was given. “This study aimed to investigate the effects of DMF on the distribution of ARGs during the AAO process in industrial WWTPs. It also examined changes in bacterial community structures and ARGs occurrence during the process, as well as the relationships between ARGs and bacterial genera, specifically pathogens. The results showed that DMF content decreased during the AAO process, while the abundance and number of ARGs and bacterial hosts increased. The study also found that the potential transfer of ARGs in pathogenic bacteria decreased after the AAO process. However, some ARGs still pose a significant threat to public health through their distribution and accumulation in non-pathogenic bacteria, which can be discharged through effluent. Further measures may be necessary to address this issue.”

6．Could the authors may be specific about their “samples replicates”?. Do sampling was carried out on different days? Please specify

Reply: Thank you for your thoughtful consideration of our manuscript. Sample duplication refers to three replicates taken at different locations at the same time. The original text "For each sample there were three independent replicates." has been changed to "Repeat sampling of each sample three times at different locations at the same time." 

7．Line 106. Please justify “The abundance of expressed ARGs”, if the analysis was DNA-based, authors can’t claim “gene expression”. In the same sense, lines 120, 129 mention transcripts. Is it correct?

Reply: Thank you for your advice, and I will revise it. Delete inappropriate statements, such as”expressed” ”transcripts”.

8．The methodology for DMF analysis is missing.

Reply: Thank you for your advice. We provide the following supplementary explanations: “The analysis was conducted using gas chromatography-mass spectrometry, a highly reliable and precise method. In order to test for the presence of DMF in the sample, it was first transferred to a suitable organic solvent and dissolved to create a DMF solution with a specific concentration. To ensure accurate results, a series of standard solutions with varying concentrations of DMF were prepared. These standard solutions covered a range of concentrations that could potentially be found in the sample under investigation. Both the sample and the standard solutions were then injected into the gas chromatograph. Careful consideration was given to the selection of the gas chromatograph column and mobile phase, ensuring optimal conditions for analysis. For qualitative identification, the mass spectrum of the DMF in the sample was compared to that of a known DMF reference. This comparison allowed for the confident identification of DMF in the sample. Quantitative analysis was carried out by comparing the peak area of DMF in the sample to that of the standard curve. By establishing the relationship between the peak area and the concentration at each point on the standard curve, a linear regression analysis was performed. This analysis enabled the accurate calculation of the DMF concentration in the sample under investigation.”

9．Table S1 is missing.

Reply: Thanks for your advice. Table S1 has been replenished.

10．How the 16S rRNA Absolute gene abundance was quantified? This methodology is missing. How do the authors explain a higher abundance in the effluent (Fig 1c)? The authors may explain better the process in the WWTP and their conditions? Does the WWTP have a settling unit for aerobic flocs? The units of 16S rRNA concentration in Figure 1c are missing.

Reply: Thank you for your thoughtful consideration of our manuscript. We provide the following supplementary detail: “While the active bacterial community was profiled by amplicon sequencing the 16S rRNA gene. To assess the diversity and relative abundance of bacteria in all samples, the V3 and V4 region of 16S rRNA was amplified .The initial enzyme activation was performed at 95 °C for 5 min, and then 35 cycles of the following program were used for amplification: 95 °C for 30 s, 58 °C for 30 s and 72 °C for 30 s. By sequenced with Illumina Hiseq 2500 platform universal primers 341F and 806R by Yuanzai biotechnology Co., Ltd (Hefei, China). The raw pair-end reads were assembled after filtering of adaptor, low-quality reads, ambiguous nucleotides, and barcodes to generate clean joined reads capturing the complete V4-V5 region of the 16S rRNA gene. Quantitative Insights Into Microbial Ecology (QIIME) was used for further data processing (3).” 

Sewage treatment plants have aerobic flocculation sedimentation units, which use aerobic bacteria and flocculants to remove organic matter and suspended matter in sewage. The aerobic flocculation sedimentation device is generally used to treat high-concentration industrial wastewater. The aerobic flocculation sedimentation device can improve the abundance of flora, because the aerobic bacteria can use the organic matter in the sewage as a carbon source and energy source to grow and reproduce. At the same time, the flocculant can increase the flocculation of the flora, making it easier for the flora to precipitate and return, so as to maintain the high concentration of flora. The increased abundance of 16S rRNA at the water outlet may be caused by the decreased flocculation effect. 16S rRNA concentration units have been added in Fig 1D

11．Line 157. Why the different IN clustering was so “obviously”?

Reply: Thank you for your thoughtful consideration of our manuscript. The original text was “In addition, the replicated samples of ANP, AEP, and EF samples were clustered while that of IN is obviously separated which indicated that the ARGs abundances were more stable in treatments than in IN.” IN is not only visibly dispersed but also exhibits a clear lack of clustering due to the diverse sources of sewage within the area. However, it is important to note that after undergoing the unified AAO treatment, a remarkable clustering phenomenon is expected to occur. The initial observation of IN's dispersion can be attributed to the varied origins of sewage within the region. Different industrial, residential, and commercial sources contribute to a wide distribution of pollutants, making it difficult to identify specific clusters. This dispersion is evident in the inconsistent patterns and irregularities observed in the sewage samples.

12．Define the MGE acronym.

Reply: Thanks for your advice. Modifications have been made in accordance with your suggestions.

13．The authors may support this discussion with a deep literature review: “We infer that the increase abundance of ARGs in WWTPs may be related to bacterium community, because many previous studies have demonstrated that some phylum of strain was closely related to the enrichment of ARGs.”

Reply: Thank you for your thoughtful consideration of our manuscript. The literature supports this conclusion.“Infer that the increase abundance of ARGs in WWTPs may be related to bacterium community, because many previous studies have demonstrated that some phylum of strain was closely related to the enrichment of ARGs (4, 5).”

14．Figure 2E, is lacking the label corresponding to each pie chart.

Reply: Thanks for your advice. The corresponding label has been added

15．How did the authors select the pathogenic bacteria from the whole bacteria community? This methodology is unclear.

Reply: Thank you for your thoughtful consideration of our manuscript. The pathogenic bacteria were selected based on solutions mentioned in previous literature (6). 

16．The authors may justify this statement “Specifically, DMF had a significant positive correlation with Bacillus, which was belonging to Firmicutes, indicated that DMF may promote the growth of Bacillus. Furthermore”. A positive correlation doesn’t mean a biological activity. This is the only discussion about the effect of DMF.. So, it seems irrelevant to the study, which contradicts the justification in the introduction. Authors may reconsider if DMF can be a determinant parameter to study.

Reply: Thank you for your thoughtful consideration of our manuscript. The positive correlation between DMF and bacillus does not directly indicate that DMF may promote the growth of Bacillus. What I want to show is that DMF may make Bacillus a competitive advantage by inhibiting the growth of other strains. Amend the original to“Specifically, DMF had a significant positive correlation with Bacillus, which was belonging to Firmicutes, indicated that DMF may make Bacillus a competitive advantage by inhibiting the growth of other strains.”

Reviewer #2: 

1 First person must not be used in the manuscript, such as our, we.

Reply: Thanks for your advice. First person has been modified.

2 In Abstract, the DMF should be first given the complete name, not only abbreviation.

Reply: Thanks for your advice. Adjustments have been made to follow your suggestions, and abbreviations are spelled out in full when they first appear.

3 the keywords were not approviate, DMF should be included.

Reply: Thanks for your advice. Keywords have been added,such as DMF.

4 The instrument of HT-qPCR should be given.

Reply: Thank you for your thoughtful consideration of our manuscript. The original text "The abundance of expressed ARGs were characterize by high throughput quantitative PCR on the Real time PCR." has been changed to "The abundance of expressed ARGs were characterize by high throughput quantitative PCR (HT-qPCR). HT-qPCR was performed using the Wafergen SmartChip Real-time PCR system (Wafergen, Fremont, CA). This SmartChip platform can be used for large-scale analysis by processing 5184-nanowell reactions per run." 

5 The gene name should be italic.

Reply: Gene names are italicized, thanks for your advice. 

6 The ARGs should be analyzed by different category.

Reply: Thank you for your thoughtful consideration of our manuscript. By analyzing 11 different ARGs, we analyzed the variation of ARGs type and abundance during AAO treatment. Sewage treatment did not change the composition of the main types of ARGs (Fig 2B-D). However, the quantity changed during treatment, for example, the inlet was mainly composed of aminoglycoside, multidrug and transposase, while the outlet was mainly composed of MGEs, multidrug and transposase (Fig 2C). In AP, ANP and AEP, multidrug, transposase and sulfanilamide are the main ARGs(Fig 2D). MexF, tna-04, and qacEdelta1-02 are specific ARGs.

7 The component of wastewater have an effect on abundance of ARGs this sentence should be changed, for “have an effect” was nonsense.

Reply: The original text "The component of wastewater have an effect on abundance of ARGs" has been changed to "The correlation between ARGs and the composition of wastewater" 

8 The effect of DMF should be focused.

Reply: Thank you for your advice. we focused the effect of DMF. “DMF, a component of industrial wastewater, poses a significant threat to both aquatic organisms and the environment. Its discharge into water bodies can lead to detrimental effects on aquatic communities(7), altering their ecological structure and function. Additionally, DMF has the potential to inhibit the growth and activity of soil microorganisms, ultimately disrupting the stability and functionality of soil ecosystems (8). Furthermore, it is important to highlight the adverse impact of DMF on human health. Prolonged exposure to high concentrations of DMF can result in various health issues, including skin and respiratory irritation, as well as liver and kidney damage (9). These potential health risks necessitate the urgent need for effective DMF removal methods. Conventional biological processing methods have proven to be ineffective in removing DMF due to its high solubility and stability. As a result, it becomes crucial to explore alternative treatment processes that can efficiently eliminate DMF from wastewater. In this regard, the AAO treatment process has emerged as a promising solution. With its capability to effectively remove DMF, the AAO treatment process offers a viable method for addressing the presence of DMF in wastewater. By employing this innovative approach, we can mitigate the negative impacts of DMF on wastewater treatment plants and ensure their optimal functioning. “

9 The effect of MGEs should be focused.

Reply: Thank you for your advice. we focused the effect of MGEs. “MGEs are fragments of DNA that can move around the genome, including transposons, integrons, and plasmids. They are capable of horizontal gene transfer between the genetic material of the bacteria, thereby rapidly spreading antibiotic resistance genes through the bacterial population (10). The spread of MGEs poses a challenge to antibiotic treatment. Because they enable bacteria to quickly acquire antibiotic resistance genes, which can cause antibiotics to become ineffective (11). In addition, because MGEs are highly plastic and adaptable, they can also promote the evolution and diversification of antibiotic resistance genes (12). This means that antibiotic use can select not only antibiotic-resistant bacteria, but also bacteria that carry MGEs, further exacerbating the antibiotic resistance problem. Therefore, understanding the effect of MGEs on antibiotic resistance genes is of great significance for formulating rational antibiotic use strategies and antibiotic resistance prevention and control.”

Reviewer #3: 

1. Authors should be aware of spelling and formatting errors, such as line 67.

Reply: Thank you for your suggestion. The adjustment has been made as per your recommendation. The original text "the abundance of ARGs in the effluents maybe high. and effluents may favor the persistence and spread of antibiotic resistance in the microbial communities of the receiving environments." has been changed to "the abundance of ARGs in the effluents maybe high. Effluents may favor the persistence and spread of antibiotic resistance in the microbial communities of the receiving environments." accordingly.

2. This study sampled only one specific water plant which is not representative and the conclusions may not be generalizable. The authors should add some related references for comparison such as Journal of Environmental Management,Volume 347, 2023, 119053, Journal of Hazardous Materials,Volume 298, 2015, Pages 303-309, Process Safety and Environmental Protection,Volume 93, 2015, Pages 68-74.

Reply: Thanks for your advice. References are used to support my conclusion (13-15).

3. COD was consistent with the amount of DMF. Is there a way to add a correlation chart and embellish Figure 1？

Reply: Thanks for your advice. We added the correlation figure Fig1C as suggested

4. Abbreviations need to be spelled out in full the first time they appear, DMF,AAO in the abstract, PCA in the methodology on line 31.MGEs in the result.in abbreviations should be placed after the corresponding full name, not at the end of each sentence.

Reply: Thanks for your advice. Adjustments have been made to follow your suggestions, and abbreviations are spelled out in full when they first appear

5. In the conclusions, it is stated that the DMFs in Figure 2b for AP, ANP, AEP, but the graph shows significantly higher values for AP than for influent and effluent.Is this due to the lack of graphs to allow for the corresponding conclusions to be drawn?

Reply: Thanks for your advice. Fig 1A shows the change of DMF content, during the treatment process, DMF content decreased significantly, with the highest IN and the lowest EF. Fig 2B is the relative copy number of ARGs and MGEs, with AP being the highest. Fig 2C is the Absolute copy number of ARGs and MGEs, the absolute abundance of ARGs and MGEs is increased. The expression in lines 159-162 is changed to ”Differentially relative abundant ARGs found in wastewater during the AAO process are displayed in Fig 2B, with AP being the highest. The application of activated sludge in AP, ANP, and AEP pool reduced the content of DMF(Fig 1A), however, it also significantly increased the absolute abundance of ARGs as well as MGEs leading to a great enrichment than IN (Fig 2C).”

1. Li J, Dijkstra P, Lu Q, Wang S, Chen S, Li D, et al. Genomics-informed insights into microbial degradation of N, N-dimethylformamide. International Biodeterioration & Biodegradation. 2021;163:105283.

2. Lu X, Wang W, Zhang L, Hu H, Xu P, Wei T, et al. Molecular Mechanism of N,N-Dimethylformamide Degradation in Methylobacterium sp. Strain DM1. Applied and environmental microbiology. 2019;85(12).

3. Caporaso JG, Kuczynski J, Stombaugh J, Bittinger K, Bushman FD, Costello EK, et al. QIIME allows analysis of high-throughput community sequencing data. Nature methods. 2010;7(5):335-6.

4. Ju F, Zhang T. Bacterial assembly and temporal dynamics in activated sludge of a full-scale municipal wastewater treatment plant. The ISME journal. 2015;9(3):683-95.

5. Jia S, Shi P, Hu Q, Li B, Zhang T, Zhang XX. Bacterial Community Shift Drives Antibiotic Resistance Promotion during Drinking Water Chlorination. Environmental science & technology. 2015;49(20):12271-9.

6. Yu Q, Feng T, Yang J, Su W, Zhou R, Wang Y, et al. Seasonal distribution of antibiotic resistance genes in the Yellow River water and tap water, and their potential transmission from water to human. Environmental pollution (Barking, Essex : 1987). 2022;292(Pt A):118304.

7. Di Mauro V, Kamyab E, Kellermann MY, Moeller M, Nietzer S, Luetjens LH, et al. Ecotoxicological Effects of Four Commonly Used Organic Solvents on the Scleractinian Coral Montipora digitata. Toxics. 2023;11(4).

8. Ul'yanovskii NV, Lakhmanov DE, Pikovskoi, II, Falev DI, Popov MS, Kozhevnikov AY, et al. Migration and transformation of 1,1-dimethylhydrazine in peat bog soil of rocket stage fall site in Russian North. The Science of the total environment. 2020;726:138483.

9. Zhang Y, Pei M, Zhang B, He Y, Zhong Y. Changes of antibiotic resistance genes and bacterial communities in the advanced biological wastewater treatment system under low selective pressure of tetracycline. Water research. 2021;207:117834.

10. Wang H, Hou L, Liu Y, Liu K, Zhang L, Huang F, et al. Horizontal and vertical gene transfer drive sediment antibiotic resistome in an urban lagoon system. Journal of environmental sciences (China). 2021;102:11-23.

11. Nguyen AQ, Vu HP, Nguyen LN, Wang Q, Djordjevic SP, Donner E, et al. Monitoring antibiotic resistance genes in wastewater treatment: Current strategies and future challenges. The Science of the total environment. 2021;783:146964.

12. Vasco K, Guevara N, Mosquera J, Zapata S, Zhang L. Characterization of the gut microbiome and resistome of Galapagos marine iguanas (Amblyrhynchus cristatus) from uninhabited islands. Animal microbiome. 2022;4(1):65.

13. Czatzkowska M, Rolbiecki D, Zaborowska M, Bernat K, Korzeniewska E, Harnisz M. The influence of combined treatment of municipal wastewater and landfill leachate on the spread of antibiotic resistance in the environment - A preliminary case study. Journal of environmental management. 2023;347:119053.

14. Huang M, Qi F, Wang J, Xu Q, Lin L. Changes of bacterial diversity and tetracycline resistance in sludge from AAO systems upon exposure to tetracycline pressure. Journal of hazardous materials. 2015;298:303-9.

15. Huang M-H, Zhang W, Liu C, Hu H-Y. Fate of trace tetracycline with resistant bacteria and resistance genes in an improved AAO wastewater treatment plant. Process Safety and Environmental Protection. 2015;93:68-74.

---

## [Decision Letter · Decision Letter 1]

7 Feb 2024

PONE-D-23-27587R1The proliferation of antibiotic resistance genes (ARGs) and microbial communities in industrial wastewater treatment plant treating N,N-dimethylformamide (DMF) by AAO processPLOS ONE

Dear Dr. Gao,

Thank you for submitting your manuscript to PLOS ONE. After careful consideration, we feel that it has merit but does not fully meet PLOS ONE’s publication criteria as it currently stands. Therefore, we invite you to submit a revised version of the manuscript that addresses the points raised during the review process.

We look forward to receiving your revised manuscript.

Kind regards,

Catarina Leite Amorim, Ph.D.

Academic Editor

PLOS ONE

Journal Requirements:

Reviewers' comments:

Reviewer's Responses to Questions

**Comments to the Author**

1. If the authors have adequately addressed your comments raised in a previous round of review and you feel that this manuscript is now acceptable for publication, you may indicate that here to bypass the “Comments to the Author” section, enter your conflict of interest statement in the “Confidential to Editor” section, and submit your "Accept" recommendation.

Reviewer #2: All comments have been addressed

Reviewer #4: All comments have been addressed

2. Is the manuscript technically sound, and do the data support the conclusions?

Reviewer #2: Yes

Reviewer #4: Yes

3. Has the statistical analysis been performed appropriately and rigorously? 

Reviewer #2: Yes

Reviewer #4: Yes

4. Have the authors made all data underlying the findings in their manuscript fully available?

Reviewer #2: Yes

Reviewer #4: Yes

5. Is the manuscript presented in an intelligible fashion and written in standard English?

Reviewer #2: Yes

Reviewer #4: Yes

6. Review Comments to the Author

Reviewer #2: (No Response)

Reviewer #4: This article presents a comprehensive analysis of the presence and distribution of Antibiotic Resistance Genes (ARGs) in industrial wastewater treatment plants (WWTPs). It successfully addresses a significant gap in the literature by focusing on industrial WWTPs, a subject that is largely overlooked in favor of studies on hospital and urban WWTPs. The methodology employed in the study, which includes the collection of 15 wastewater samples from five stages of the anaerobic anoxic aerobic (AAO) process, is robust and lends credibility to the findings.

The article is well-structured and logically organized, with each step of the research process clearly delineated. It does an excellent job of presenting complex scientific data in a readable and understandable format. The findings are clearly articulated, and the use of graphs and tables to depict data enhances readability and understanding. The study's findings are significant and troubling, revealing a clear increase in ARGs in effluents of biological treatments and suggesting that the AAO process may serve as a microbial source, increasing ARGs' total abundance. The article also provides an intriguing insight into the connection between the structure of bacterial communities and the dynamics of ARGs. Overall, this is an excellent and timely piece of research that contributes significantly to our understanding of ARGs in industrial WWTPs. It is well-executed, thorough, and provides a strong foundation for future research in this area.

The results of the study are quite comprehensive and detailed. However, there are several areas that could benefit from some fine-tuning and adjustments. I think that the paper is publishable in the journal after some minor revision. See my specific comments below:

1. Providing a graphical abstract can aid readers in better understanding the article.

2. The current text seems to include conclusions within the results section. Typically, it is best to keep these sections separate, with the results section strictly presenting the findings, and interpretation or implications discussed in the conclusion section.

3. There are several typos or grammatical errors that need to be corrected.

4. The writing style is quite passive. In scientific writing, active voice is often preferred as it is more direct and concise.

5. Line 106, "The wastewater was collected from the Shen Nuobei industrial wastewater treatment plant (118.507° N, 31.689° E)，Ma’anshan city, Anhui province, China, in December 2021.", suggest to change to "In December 2021, wastewater was sampled from the Shen Nuobei industrial wastewater treatment plant located in Ma'anshan city, Anhui province, China (118.507° N, 31.689° E)."

6. Line 110, "Collected 2 L of wastewater samples from influent, anaerobic, anoxic, aerobic, and effluent tanks.", suggest to change to "Two liters of wastewater samples were collected from the influent, anaerobic, anoxic, aerobic, and effluent tanks."

7. Line 111, "Repeat sampling of each sample three times at different locations at the same time.", suggest to change to "Each sample was collected three times from different locations concurrently."

8. Line 301, "ARGs has increased rapidly" should be changed to "ARGs have increased rapidly".

9. Line 303, "a previous research" should be changed to "previous research".

10. Line 305, "lead to a significant increase" should be changed to "leading to a significant increase".

11. Line 341, "withour" should be corrected to "with our".

12. Line 380, "the horizontal gene transfer via natural transformation between nonpathogen is still high" should be changed to "the horizontal gene transfer via natural transformation among non-pathogens remains high".

13. Line 382, "implementing measure" should be changed to "implementing measures".

7. PLOS authors have the option to publish the peer review history of their article (what does this mean?). If published, this will include your full peer review and any attached files.

Reviewer #2: No

Reviewer #4: No

---

## [Author Response · Author response to Decision Letter 1]

10 Feb 2024

Dear Reviewer,

Thank you for taking the time to provide such a thorough and constructive review of our manuscript. We appreciate your positive comments about the comprehensive and detailed results of our study and are pleased that you found our research to be significant and timely. We also acknowledge your suggestions for improvement and have made the necessary revisions as per your comments.

1. Providing a graphical abstract can aid readers in better understanding the article.

Reply: Thank you for your advice. We have included a graphical abstract to help readers better understand the article's content. In line 112, we add the sentence“The experimental analysis process is shown in the graphical abstract (fig. S1).”

2. The current text seems to include conclusions within the results section. Typically, it is best to keep these sections separate, with the results section strictly presenting the findings, and interpretation or implications discussed in the conclusion section.

Reply: Thank you for your advice. We agree with your suggestion and have now separated the results and conclusion sections. The conclusions have been moved to a dedicated section where we discuss the interpretation and implications of our findings. We move the sentence “AAO process can remove the DMF efficiently and bacteria plays an important role in this process.” from result to conclusion.

3. There are several typos or grammatical errors that need to be corrected.

Reply: We apologize for any confusion that may have caused. We check the article again. All typos and grammatical errors have been corrected. 

4. The writing style is quite passive. In scientific writing, active voice is often preferred as it is more direct and concise.

Reply: Thank you for your advice. We have revised the manuscript to incorporate a more active voice, making it more direct and concise.

5. Line 106, "The wastewater was collected from the Shen Nuobei industrial wastewater treatment plant (118.507° N, 31.689° E)，Ma’anshan city, Anhui province, China, in December 2021.", suggest to change to "In December 2021, wastewater was sampled from the Shen Nuobei industrial wastewater treatment plant located in Ma'anshan city, Anhui province, China (118.507° N, 31.689° E)."

Reply: Thank you for your advice. The sentence on Line 106 has been revised as suggested.

6. Line 110, "Collected 2 L of wastewater samples from influent, anaerobic, anoxic, aerobic, and effluent tanks.", suggest to change to "Two liters of wastewater samples were collected from the influent, anaerobic, anoxic, aerobic, and effluent tanks."

Reply: Thank you for your advice. The sentence on Line 110 has been revised as per your suggestion.

7. Line 111, "Repeat sampling of each sample three times at different locations at the same time.", suggest to change to "Each sample was collected three times from different locations concurrently."

Reply: Thank you for your advice. The sentence on Line 111 has been revised to clarify that each sample was collected three times from different locations concurrently.

8. Line 301, "ARGs has increased rapidly" should be changed to "ARGs have increased rapidly".

Reply: Thank you for your advice. The sentence on Line 301 has been corrected to "ARGs have increased rapidly".

9. Line 303, "a previous research" should be changed to "previous research".

Reply: Thank you for your advice. The phrase "a previous research" on Line 303 has been corrected to "previous research".

10. Line 305, "lead to a significant increase" should be changed to "leading to a significant increase".

Reply: Thank you for your advice. The phrase "lead to a significant increase" on Line 305 has been corrected to "leading to a significant increase".

11. Line 341, "withour" should be corrected to "with our".

Reply: Thank you for your advice. The typo "withour" on Line 341 has been corrected to "with our".

12. Line 380, "the horizontal gene transfer via natural transformation between nonpathogen is still high" should be changed to "the horizontal gene transfer via natural transformation among non-pathogens remains high".

Reply: Thank you for your advice. The sentence on Line 380 has been revised as suggested.

13. Line 382, "implementing measure" should be changed to "implementing measures".

Reply: The phrase "implementing measure" on Line 382 has been corrected to "implementing measures".

Once again, we thank you for your valuable feedback. We believe that these revisions have significantly improved the manuscript and hope that it is now suitable for publication in the journal.

Best Regards,

Xuan Gao

---

## [Decision Letter · Decision Letter 2]

16 Feb 2024

The proliferation of antibiotic resistance genes (ARGs) and microbial communities in industrial wastewater treatment plant treating N,N-dimethylformamide (DMF) by AAO process

PONE-D-23-27587R2

Dear Dr. Gao,

We’re pleased to inform you that your manuscript has been judged scientifically suitable for publication and will be formally accepted for publication once it meets all outstanding technical requirements.

Kind regards,

Catarina Leite Amorim, Ph.D.

Academic Editor

PLOS ONE

Reviewers' comments:

Reviewer's Responses to Questions

**Comments to the Author**

1. If the authors have adequately addressed your comments raised in a previous round of review and you feel that this manuscript is now acceptable for publication, you may indicate that here to bypass the “Comments to the Author” section, enter your conflict of interest statement in the “Confidential to Editor” section, and submit your "Accept" recommendation.

Reviewer #4: All comments have been addressed

2. Is the manuscript technically sound, and do the data support the conclusions?

Reviewer #4: Yes

3. Has the statistical analysis been performed appropriately and rigorously? 

Reviewer #4: Yes

4. Have the authors made all data underlying the findings in their manuscript fully available?

Reviewer #4: Yes

5. Is the manuscript presented in an intelligible fashion and written in standard English?

Reviewer #4: Yes

6. Review Comments to the Author

Reviewer #4: The authors have revised the paper carefully according to the comments, and addressed all the comments raised by me, I think the revised paper can be accepted for publication in its present form now.

7. PLOS authors have the option to publish the peer review history of their article (what does this mean?). If published, this will include your full peer review and any attached files.

Reviewer #4: No

---

## [Editor Report · Acceptance letter]

18 Mar 2024

PONE-D-23-27587R2 

PLOS ONE

Dear Dr. Gao, 

I'm pleased to inform you that your manuscript has been deemed suitable for publication in PLOS ONE. Congratulations! Your manuscript is now being handed over to our production team.

Kind regards, 

on behalf of

Dr. Catarina Leite Amorim 

Academic Editor

PLOS ONE